# Hidden and Understaffed: Exploring Canadian Medical Laboratory Technologists’ Pandemic Stressors and Lessons Learned

**DOI:** 10.3390/healthcare11202736

**Published:** 2023-10-14

**Authors:** Patricia Nicole Dignos, Ayesha Khan, Michael Gardiner-Davis, Andrew Papadopoulos, Behdin Nowrouzi-Kia, Myuri Sivanthan, Basem Gohar

**Affiliations:** 1Department of Population Medicine, University of Guelph, Guelph, ON N1G 2W1, Canada; 2Department of Occupational Science and Occupational Therapy, University of Toronto, Toronto, ON M5G 1V7, Canada; 3Centre for Research in Occupational Safety & Health, Laurentian University, Sudbury, ON P3E 2C6, Canada

**Keywords:** medical laboratory technologists, COVID-19, Canada, stressors, patient care, mental health, qualitative

## Abstract

(1) Background: The COVID-19 pandemic has highlighted the critical role of medical laboratory technologists (MLTs) in the healthcare system. Little is known about the challenges MLTs faced in keeping up with the unprecedented demands posed by the pandemic, which contributed to the notable staff shortage in the profession. This study aims to identify and understand the stressors of MLTs in Canada and the lessons learned through their lived experiences during the pandemic. (2) Methods: In this descriptive qualitative study, we conducted five semi-structured focus groups with MLTs working during the pandemic. The focus group sessions were audio-recorded and then transcribed verbatim. Thematic analysis was used to inductively code data and identify themes. (3) Results: A total of 27 MLTs across Canada participated in the study. Findings highlighted four key themes: (i) unexpected challenges navigating through the uncertainties of an ever-evolving pandemic; (ii) implications of staff shortage for the well-being of MLTs and quality of patient care; (iii) revealing the realities of the hidden, yet indispensable role of MLTs in predominantly non-patient-facing roles; and (iv) leveraging insights from the COVID-19 pandemic to enhance healthcare practices and preparedness. (4) Conclusion: The study provides in-depth insight into the experiences of MLTs across Canada during the pandemic. Based on our findings, we provide recommendations to enhance the sustainability of the laboratory workforce and ensure preparedness and resiliency among MLTs for future public health emergencies, as well as considerations as to combating the critical staff shortage.

## 1. Introduction

Coronavirus disease (COVID-19) brought unprecedented challenges to the healthcare system. It placed a significant burden on healthcare professionals working on the front lines, increasing workplace demands on healthcare professionals, along with the heightened risk of exposure to the virus [1,2]. In a recent survey of Canada’s healthcare workers during the COVID-19 pandemic, most healthcare workers (95.0%) reported that the pandemic impacted their job, and a large majority (86.5%) reported increased work stress during the pandemic compared to before the pandemic [3]. Notably, healthcare workers were already facing work-related stress as well as exhibiting elevated rates of burnout and emotional exhaustion before the pandemic [4,5]. However, the advent of the pandemic markedly increased the level of stress among healthcare workers [5,6,7]. Studies have consistently shown that the pandemic had a negative psychological impact on healthcare workers working during the pandemic, revealing high levels of burnout, anxiety, emotional exhaustion, depression, and post-traumatic stress [5,8,9,10,11,12,13].

Among these healthcare professionals are medical laboratory technologists (MLTs), who work primarily behind the scenes. MLTs conduct medical laboratory testing and analyses to assist in the diagnosis, treatment, monitoring, and prevention of disease [14]. MLTs work in various clinical laboratories, including hospitals, blood banks, public and private clinics, and research institutions. They practice in various specialized areas, including clinical chemistry, hematology, transfusion medicine/science, diagnostic cytology, cytogenetics, histology, immunology, microbiology, bacteriology, and virology [15]. 

The pandemic highlighted the critical role of MLTs, underscoring the importance of their work in the healthcare system. Their expertise facilitated the surveillance of the virus and supported public health efforts to prevent its spread [16]. However, the unparalleled challenges brought by the pandemic placed considerable strain on MLTs, especially their ability to keep up with the demands [17]. This struggle was exacerbated by the shortage of MLTs in Canada. According to the Canadian Society for Medical Laboratory Science (CSMLS), Canada is facing a serious shortage of MLTs. MLT shortage was an existing issue prior to the pandemic but was further accelerated with the onset of the pandemic [18]. Recent data from the Canadian Institute of Health Information (CIHI) in 2021 shows that there were a total of 19,643 MLTs across Canada, with females making up nearly 80% of the workforce and approximately three-quarters of MLTs falling within the age range of 30 to 59 years old [19]. With the aging population of MLTs, it is projected that half of the MLTs will be retiring in the next few years, leaving a gap in the workforce [20]. With the emergence of the virus, the surge of testing demands increased the workload of MLTs. Before the onset of the COVID-19 pandemic, medical laboratory professionals conducted over 440 million tests annually in Canada [18]. The arrival of the pandemic imposed increased demands on the laboratory system, with the addition of hundreds of thousands of COVID-19 tests daily. These increased demands of the pandemic, coupled with the staff shortage, had a detrimental effect on the well-being of MLTs [21]. Compared to pre-pandemic timespans, MLTs experienced increased levels of burnout during the pandemic [22]. Furthermore, they experienced higher job dissatisfaction during the pandemic than before the pandemic [23]. 

The impact of the pandemic has been extensively studied as to visible healthcare workers, such as physicians and nurses [1,6,24]. However, there is limited research examining the impact of the pandemic on MLTs and the unique challenges they face. A provincial study examining pandemic-related stressors of medical laboratory professionals in Ontario has been conducted previously [25]. The study’s findings revealed medical laboratory professionals are often forgotten compared to other healthcare providers, and that the pandemic contributed to the existing staffing shortage and exacerbated their poor working conditions. To date, however, little is known about the experiences of MLTs across the country during the pandemic.

This qualitative study aims (1) to identify and understand the stressors of MLTs working during the pandemic and (2) to identify the lessons learned from the COVID-19 pandemic, with emphasis on well-being and how to optimize healthcare delivery. The insights gained from this study can contribute to promoting resilience among MLTs and address existing gaps in the laboratory workforce to ensure workforce sustainability and enhance the preparedness of the healthcare system for future public health emergencies. 

## 2. Methods

### 2.1. Study Design

The present study explores the stressors of MLTs and lessons learned on a national scale, building upon previous work by authors B.G. and B.N.-K. on the same population within a provincial context [25]. For this qualitative, descriptive study, we held focus groups with MLTs from provinces across Canada. We used focus groups to gain in-depth insight into participants’ lived experiences and encourage interactive discussion in the group. Focus groups provide rich, detailed descriptions of the knowledge, attitudes, and experiences of participants through group interaction. The interactional nature of focus groups allows participants to engage in dialogue by exchanging ideas, building upon each other’s thoughts, and expanding on or contradicting each other’s responses [26].

The focus groups were conducted virtually through Microsoft Teams Version 22.08.1 [27] from July 2022 to January 2023. Virtual focus groups allow flexibility, accessibility, and the ability to recruit geographically diverse participants, including participants from rural and hard-to-reach areas [28,29]. Five focus groups were held, each approximately 90 min in length. Participants were assigned to focus groups based on the participants’ availability, with efforts made to include individuals from different provinces to enhance the diversity of the discussions. The focus groups were moderated by M.S., who was assisted by P.D., and supervised by B.G. Combined, the research team has expertise in public health, psychology, and applying qualitative methodologies with healthcare groups. 

### 2.2. Participant Recruitment

Eligible participants were medical laboratory technologists in Canada working during the pandemic. Participants were identified through a multi-faceted approach. We partnered with various associations and regulatory bodies across Canada, including the CSMLS, the Medical Laboratory Professionals’ Association of Ontario, the British Columbia Society of Laboratory Science, the College of Medical Laboratory Technologists of Alberta, the College of Medical Laboratory Technologists of Manitoba, the Nova Scotia College of Medical Laboratory Technologists, and the Newfoundland and Labrador College of Medical Laboratory Science to help facilitate the recruitment process. Participants were also recruited using social media platforms (i.e., LinkedIn, Twitter, and Facebook). In addition, a snowball sampling technique was applied by asking existing participants to assist with recruiting potential participants. To minimize the risk of fraudulence, participants’ names were verified through the public registry of their respective provinces before participating. Participants received a $50.00 Amazon gift card for their participation in the study.

### 2.3. Ethical Considerations

We obtained ethics approval from the University of Guelph’s Research Ethics Board (22-03-001). Before participating in the study, participants were informed about the purpose, risks, benefits, confidentiality of information, anonymity, and voluntariness of the study. Written informed consent was obtained from each participant. To maintain privacy and confidentiality, participants were asked to refrain from using any identifying information that others might recognize. Participants were reminded that the session would be audio-recorded for the accuracy of data collection.

### 2.4. Data Collection

A semi-structured interviewing method was applied, guiding the discussions with open-ended questions (Appendix A). Follow-up questions were also added as necessary. We began the focus group sessions by asking participants demographic questions, including their age, gender, years of practice, work experience, region, occupational setting, and whether they work in a rural or urban location. Next, we asked the participants about their experiences and stressors during the pandemic. We also asked about their major pre-pandemic stressors for context. After this, participants were asked about the implications of these stressors for patient care. To reduce occupational stressors in the event of future pandemics or public health emergencies, we also asked participants about the lessons learned during the pandemic. Finally, we asked the participants how to optimize the healthcare system, from the lens of their profession as MLTs. Throughout the focus groups, we restated and paraphrased the participants’ responses as a form of member-checking (or participant validation) to ensure the credibility and trustworthiness of the data. Open coding was applied throughout data collection, allowing for continuous comparison of new data with existing information. This iterative approach helped in determining data saturation by detecting redundancy in responses without introducing new insights. Data saturation was reached after conducting five focus groups. 

### 2.5. Data Analysis

The focus group recordings were transcribed verbatim. To ensure accuracy and improve credibility, two researchers (P.D. and M.G.-D.) reviewed the transcripts while listening to the audio recording. Data analysis was guided by Braun and Clarke’s six steps of thematic analysis [30], using Quirkos [31], a qualitative data analysis software, to code the textual data. To collect rich, context-specific data and gain deeper insights into the unique stressors experienced by MLTs during the pandemic, two authors (P.D. and A.K.) analyzed the transcripts and inductively coded the data. Given the limited information in the existing literature, an inductive approach was applied, allowing patterns to emerge from the data without imposing preconceptions or theoretical frameworks. Open coding was used to generate initial codes, and related codes were then organized into meaningful categories based on similarities. Two additional researchers (B.G. and M.G.-D.) reviewed the codebook to further enhance reliability. P.D. and A.K. further analyzed the categories to identify recurring patterns and develop themes and subthemes, which were later reviewed by the team. Selected excerpts were added to the results to further illustrate the findings, enhancing transparency and reader engagement. To ensure the quality of our study, the consolidated criteria for reporting qualitative research (COREQ) was utilized (Appendix A) [32].

## 3. Results

### 3.1. Participant Demographic

We reached data saturation after conducting five focus groups. A total of 27 medical laboratory technologists across Canada participated in the study, with 22 participants identifying as women and five participants identifying as men (Table 1). The participants’ ages ranged between 23 and 55 years, with a mean age of 37 (SD = 9.3). Among the participants, nine were from Ontario, seven were from British Columbia, seven were from Alberta, one was from Manitoba, two were from Nova Scotia, and one was from Newfoundland and Labrador. Participants worked in rural and urban locations in various clinical settings, including hospitals, private laboratories, provincial laboratories, clinics, community centers, non-profit organizations, and manufacturing/distribution. Participants had varying lengths of experience in their respective practices.

Four key themes emerged from our data: (1) unexpected challenges navigating through the uncertainties of an ever-evolving pandemic; (2) implications of staff shortage for the well-being of MLTs and the quality of patient care; (3) revealing the realities of the hidden, yet indispensable role of MLTs in predominantly non-patient-facing roles; and (4) leveraging insights from the COVID-19 pandemic to enhance healthcare practices and preparedness (Table 2). Figure 1 shows the thematic map outlining the link between themes and subthemes.

### 3.2. Unexpected Challenges Navigating through the Uncertainties of an Ever-Evolving Pandemic

#### 3.2.1. Fear of the Unknown and Contagiousness of the Virus

Our findings revealed that MLTs experienced fear and anxiety due to the uncertainty of the pandemic. Participants defined the fear of the unknown as a significant contributor to the stress which they had experienced, especially at the beginning of the pandemic, due to the limited knowledge and understanding of the COVID-19 virus. 


*FG-80: “So, I found the, the biggest stressor for me at least, like for, for the first wave, when it first started, was the unknown. So, you sort of like, everyone in the public had we don’t know what the effect of this is, we don’t know how contagious it is. We don’t really know what’s going on.”*



*FG-89: “Uh, I think in the beginning it was, it was really the uncertainty of the pandemic like, um, yeah, just you don’t know what’s gonna happen next.”*


The fear of infection emerged as a prominent stressor during the pandemic. Due to the high-risk nature of their work, which involves handling potentially infectious patient samples, participants expressed their profound fear of contracting the virus. Some participants whose duties included phlebotomy and sample collection expressed heightened concerns about contracting the virus. The risk of exposure to COVID-19 created a sense of unease and concern among participants for their own personal health and safety. This fear of infection felt by MLTs was intensified as they found themselves ill-equipped to work safely due to the shortage of personal protective equipment (PPE) during the early stages of the pandemic, which increased their susceptibility to the virus. Participants described the manner in which the lack of adequate PPE compromised their safety and their ability to safeguard their well-being, which added a layer of stress to their already-demanding jobs. 


*FG-104: “…just anxiety about catching COVID and going to work and having to put myself at risk while this maximum workload was still being placed upon us, so like my own health and safety.”*



*FG-79: “…there was a shortage of masks and we wanted to reserve the masks to the nurses and doctors or other professionals who interacted with the patients directly.”*


Participants’ concerns extended beyond the workplace. A cogent stressor among participants was the fear of infecting their family members, including children and those who were immunocompromised. During the initial wave of the COVID-19 pandemic, the pervasive lack of comprehensive information about the transmission of the virus was a constant source of fear and distress. Participants described the ways in which the stress of not knowing how to protect themselves and their families heightened their anxiety. This fear was particularly pronounced for participants with young children and those who lived with family members who were more vulnerable and at higher risk from COVID-19 infection, such as elderly parents and immunocompromised family members. Participants described taking strict precautions, such as disinfecting personal belongings, showering, and changing clothes immediately upon returning home from work before interacting with their families, demonstrating the lengths they went to minimize the risk of transmission.


*FG-53: “…in the first wave and especially at the beginning was, was the fear of infection and the fear of, worse, the fear of bringing it home. I, I, I don’t know many people who in, in health care, who weren’t experiencing that, the fear of bringing it home… especially in the beginning when we didn’t have enough information about transmission and modes of contact and means of managing containment. We didn’t have any of that.”*



*FG-85: “Like for me, I had a brand-new baby at home and I had a, a child in school and coming home and not like, trying not to bring it home. I was like scrubbing down and showering, going straight down, stripping my scrubs off and putting them in the, um, wash… And so, then I was changing at work and then coming home and so, it was a lot of that stress and just not knowing, and trying to keep your, your family safe. Even knowing that you’re, you’ve just walked into a, a positive room and like, “How do I not bring this home to my brand-new baby? How do I not bring this home to my, my husband and, and little son?”*



*FG-80: “My husband is also exposed to patients and, and my kids go to school so, there was also, you know, that. And I have, you know, my mom’s in her 70s and we’re worried about her and, you know that kind of stuff?… And you know, when you go to work and then you worry about bringing it home.”*



*FG-82: “I also live with someone who’s immunocompromised, and our lab was right beside the intensive care unit, so I was a bit concerned about bringing anything home.”*


Participants also expressed their concerns about inadvertently infecting patients, recognizing the far-reaching consequences and the potential implications for immunocompromised patients, including vulnerable populations. They acknowledged their responsibility as healthcare workers to adhere to infection and prevention-control measures to ensure the well-being of their patients. 


*FG-79: “Because we didn’t know much about COVID such that how it’s spread, etcetera, I was hoping that I will, I was not spreading the virus to patients by coming to work.”*


#### 3.2.2. Adapting to an Ever-Changing and Unpredictable Environment

The rapidly evolving nature of the pandemic contributed to the stress and anxiety felt by participants. Participants described the unpredictability at work during the pandemic, when each day brought uncertainties and new challenges, which caused ambiguity in navigating the workplace. Due to the rapid development of knowledge about the virus, MLTs were tasked to adapt to frequently modified health and safety protocols. These changes included the use of PPE, infection prevention, and control measures. Participants noted that the changing protocols and inconsistent communication caused confusion among MLTs. As an added stressor, management expectations changed rapidly, causing a sense of ambiguity as to their roles and instability in their work routines, thus elevating their stress levels. Participants also highlighted the challenge of keeping up with the rapid advancements, which included learning new testing methods and novel technologies.


*FG-80: “And then just every time we went to work, there was a change. So, one day it would be like, okay, well, you have to wear a mask all the time, well, you do this, and now it’s you have to wear an N-95 and now it’s, okay well, we want you to wear an N-95 and a face shield and then, oh, well, you don’t really need any of that. So, every day when we went to work, there was something new, there was a new procedure to learn, there was a new protocol to follow, and it’s just, we sort of had these, like, debriefings every day.”*



*FG-97: “Um, I will say for us one of the biggest stressors would be a heavy workload which increased exponentially for us, especially during COVID because our lab was very much involved in the COVID testing. And with that, the need to adapt to change very quickly and then like, to learn new processes and new testing methods in a very short period of time.”*



*FG-11: “…I found during COVID there was a need for a lot of, um, adaptability and there was a lot of changing just kind of happening in a very short time scale and a lot of lab[s] just weren’t prepared to take that on.”*


#### 3.2.3. Resource Constraints and Supply-Chain Challenges

The pandemic also presented challenges to resource availability. Specifically, the pandemic created a surge in demand for testing supplies, causing disruptions and delays to the supply chain, which significantly strained the availability of these resources and impacted the efficiency of laboratory operations. Participants described situations in which the shortages of essential supplies and resources affected laboratory operations, further contributing to their stress as managements’ demands remained high. Participants expressed the belief that due to the supply shortages, they had to be resourceful by using the resources available to them and exploring alternative options. Consequently, this increased the time required to complete their tasks, consequently adding to their workload. Participants also described the fact that much of their equipment had become outdated or broken and needed replacement. The limited availability of the necessary equipment hindered the quality and timeliness of laboratory testing processes. These challenges further contributed to the burdens experienced by MLTs.


*FG-105: “I just wanted to also add the strain of supply chain, um, where our current, like most common supplies are often running out with no alternative to replace. Um, and for the lab, it means a lot because we actually have to validate new reagent, new test tubes. And this is just increasing workload to already stretched workforce, um, and we’ve never seen this in the past.”*



*FG-96: “And then the final stressor that I thought of that was touched on was just supply issues, but not only COVID-specific supplies but even things like our gold top tubes. At one point, we were very near to the point of sending our patients to another location because we had to save the very few (saline) tubes that we had in case we had stats, in case we had emergencies, that sort of thing, um, because we just could not source it. We’re a small clinic. We were not on the top priority list for the supplier. So, it took a lot to be able to get a hold of the supplies that we needed, even for things that weren’t COVID-related.”*



*FG-107: “There’s a lack of equipment required to do our, our job with, you know, sometimes a lot of the instruments are getting older.”*


### 3.3. Implications of Staff Shortage for the Well-Being of MLTs and the Quality of Patient Care 

#### 3.3.1. Burnout due to Increased Workload and Prolonged Hours Leading to Attrition 

The pandemic increased testing demands, significantly increasing the work volume of MLTs. This placed a significant strain on MLTs, as they were tasked to process a larger number of samples in a shorter time. Participants reported working longer hours, sometimes foregoing breaks, to keep up with the escalating demands. This strain was further compounded by a notable staff shortage that existed before the pandemic. The increased workload and prolonged hours along with the limited human resources exacted a toll on the physical, mental, and emotional well-being of MLTs. Unsurprisingly, this work environment led to a notable rise in workplace injuries and an increased incidence of sick leaves and absences.


*FG-104: “…you know, you had three night shifts, then one day off then had to work seven day shifts in a row and you never got to see your spouse or you had to work stat holidays.”*



*FG-94: “There’s been injuries at work and with how we have staff shortages and increased load, people are trying to push harder, work faster so those injuries are happening more often.”*



*FG-107: “It was a, a big stress on me because I, I needed to get everything up and running because people’s lives were, um, on the line. So, that was, it was so bad that I actually ended up in the hospital with a cardiac event. And stupid me thought, okay, well, uh, I should just go back, I’m okay now, I’m gonna go back. So, I tried for a few more months and the stress, it just didn’t, it didn’t stop, like it just kept actually rising. So, I ended up going on medical leave.”*


The overwhelming workplace pressure and insufficient staffing led to heightened levels of stress, which led to retention challenges. The cumulative stress resulted in burnout among many MLTs, causing them to leave the profession or retire early, exacerbating the staff shortage. As a result, this substantially increased the workload for the remaining MLTs.


*FG-89: “Like my experience in my workplace was that a lot of the staff burnt out during the pandemic… Uh, our sample load increased dramatically and there was a lot of people that weren’t adjusted to that to, um, that kind of workload…they ended up retiring early because they couldn’t take the, they couldn’t take the workload… because it was just, like it was just too much work. And then so that, that caused our staff to be in a little bit of a crisis because suddenly there was many people leaving our lab.”*



*FG-104: “Before COVID, we didn’t necessarily ever have staffing shortages because it would be like one tech got hired and one tech would retire and it was sort of this ongoing trade off and it wasn’t too bad, but we found with COVID we had a lot of techs, uh, for about halfway through 2021, we had a lot of retirees and a lot of people leaving permanently due to stress leave, so we were really running out of people because so many people were leaving.”*


The increased workload and limited staffing imposed on the remaining staff had a negative impact on the overall job satisfaction and well-being of MLTs. As a result, some participants expressed intentions to leave the profession and consider alternative career paths at some point during the pandemic. Also, some participants revealed that the workload pressure led them to make the decision to leave their previous workplace and seek employment in less-demanding settings.


*FG-105: “For me, I often had to re-evaluate if I wanted to retire because I’m actually at an age where I can retire. Uh, so I did have quite a few times where I would just wanted to give up because on a daily basis, I just deal with problems, like everyone was so negative, and problem-solving all the time and having to find all the solutions.”*



*FG-89: “I actually like reconsidered my profession many, many times. I would think it was such a mistake to get into healthcare, that I just wanted out, that I wouldn’t recommend this health -, this profession to anyone…”*



*FG-51: “Even before the pandemic. I think we had maybe half a year or a year when I first started where we were fully staffed, and it was, uh, sort of a steady decline from there on out and it got much, much worse to the point where I actually left that lab and moved to a different lab because it was too short staffed and the demands were still the same or a little bit more, and it ended up putting a little too much strain on people who are left.”*


Participants explained that senior MLTs who retired during the pandemic were replaced by newer and less-experienced staff. This increased the burden on the remaining staff members, as they had the added responsibility of training the new MLTs in addition to their existing workload, further straining an already-stretched workforce. Participants noted that new MLTs found themselves confronted with an overwhelming and highly stressful work environment, leaving them unable to handle the pressure. As a result, some newly recruited MLTs left the profession. Reportedly, while facing a shortage of qualified MLTs, the turnover of newly trained staff worsened the situation, making it increasingly difficult to fill the workforce gap.


*FG-89: “…since we had, um, uh, since it was so stressful at our lab, people would come and be trained and then leave immediately after finding a new job just because, just the workload that they were thrown into was so, was so high that they just looked for, for a new job.”*



*FG-49: “…it’s stressful and I can’t imagine being a new person and having expectations … You don’t wanna make mistakes and there’s a lot of pressure to make sure that what you’re putting out is accurate and correct and, on both sides, I think from our new hires, and that’s why we’ve actually had a lot of new hires leave because it’s so chaotic, they, they just can’t take the environment.”*


#### 3.3.2. Increased Testing Errors due to Workload Pressures and Inadequate Training Compromising Patient Care

Participants reported increased rates of laboratory errors due to the high influx of testing. They explained that the pressure to meet demands and turnaround times led them to sacrifice the rigor and quality of their work. Participants reported instances in which they were instructed to ‘cut corners’ to expedite the testing process. However, participants expressed their concerns about these shortcuts in testing procedures which could potentially compromise the quality and integrity of test results and, in turn, pose risks to patient-safety outcomes. Deviating from standard protocols weighed on the participants, as they felt that their professional integrity was being compromised.


*FG-39: “I feel that when you are constantly under stress, sometimes I, I might have and I, I might have and so many other MLT’s when they were under so much stress, they might have accidentally made any transcriptional errors in the reporting the critical results. Some results might have been missed to call, which could affect the patient care, because those patients are waiting for those results so that they, their, their treatment could begin and stuff like that, but there were there were situation where the sample was too old to be loaded onto the analyzer. But we were told to cut corners and not to reject the specimen and just continue processing them, which directly implements the patient care. Because if the patient was actually a positive but now that the specimen integrity has lost, that patient might come out as a negative. So, there, there were so many cut corners from the management.”*


Participants also highlighted the fact that the lack of training contributed to the occurrence of errors. As new protocols were rapidly introduced and implemented during the pandemic, they had to quickly learn and adapt to the changes to meet the high testing demands. Additionally, there were reportedly more errors due to inadequate training for newly hired MLTs during the pandemic. Participants explained that due to the limited staffing, there was reduced capacity to provide proper training for new MLTs. The lack of proper training and mentorship for new MLTs reportedly contributed to an increased likelihood of errors.. Correcting these errors was a time-consuming and labor-intensive process, as they often required recollection, retesting, and reanalysis of samples, which further compounded the workload of MLTs. Participants reported that these errors and the subsequent corrective measures caused delays in diagnosis and treatment, impacting patient care and health outcomes.


*FG-95: “Just to add on, I want to say that I feel that, that training, the training period was inadequate for a lot of the processes that were being implemented just because due to COVID, it needs to be implemented very fast and very soon. So, not only is the training inadequate, but staff wasn’t- staff weren’t sure what they were doing and there’s a lot of corrective reports that went out. So, I think it added on to a lot of stress that was in my workforce…”*



*FG-98: “So, if we are, yeah, um, so if we’re putting out like grads who are not fully trained or they may not know certain things, then when we put them out to the workforce and they have the, the employer has these expectations like you have to x, y, z and they don’t know those things, it may require like a recollection from a patient if they collect the wrong tube, whether that be like blood or if it’s a microbiology sample if they plate the, if they inoculate the plates incorrectly, stuff like that. And we have seen cases like this pop up, so it requires a recollection from a patient, which will just ultimately delay patient care.”*



*FG-95: “Uh, I know that there was a lot more mistakes that were made. And then, there was a lot of corrective reports just because of the burnout and it was just small mistakes that were missed that ended up being reported out. And then we’d have to phone and correct it. So, there was implications at the end of it.”*


#### 3.3.3. Moral Distress and Guilt Faced by MLTs during the Pandemic

Our findings revealed MLTs experienced moral distress and guilt as they worked during the pandemic. Participants were placed in morally distressing situations, in which their test results could have potentially affected life-saving decisions. They understood the immense responsibility that came with their work and expressed the emotional weight of knowing that their actions had direct consequences for patient outcomes. Participants also reported that because of the high priority and large volume of COVID-19 testing, other tests, such as cancer testing, were delayed. These delays added to the moral burden experienced by MLTs as they were aware that this could potentially delay diagnosis and treatment for patients. 


*FG-53: “And so, you ended up with this backlog or you were having conversations about whether or not to do this test this month and whether or not the patient could wait a month or the next month for the test and there was conversations that should never be had that were being had and it left me as a professional feeling inadequate, which I think really wore on all of us. And that was through start to finish. There wasn’t a time where that wasn’t a problem.”*



*FG-53: “…during the 2nd and the 3rd wave, some of the testing panels we were performing in my particular area were looked at as a possible means of determining who would be ventilated and who would not be ventilated, in the case where a lack of ventilators became possible. And both in the 2nd and the 3rd waves, those conversations were definitely happening. And knowing that the numbers you were generating would literally determine the fate of a human being, was incredibly stressful, way more than I had ever signed up for in my career.”*


The findings also revealed that MLTs often experienced feelings of guilt. Participants expressed feelings of guilt for taking breaks, because they felt a sense of obligation to deliver timely results. This guilt was further intensified by the high-pressure work environment, where they constantly strove to meet demanding deadlines. Participants expressed reluctance to leave unfinished work for fear of overburdening their coworkers. Participants also reported working unpaid overtime to ensure that tasks were completed before passing them on to their colleagues.


*FG-39: “We also needed a break but you can’t. You badly want to go on your coffee break, but when you actually go on your coffee break, you cannot enjoy your coffee break, because now you’re constantly thinking about going back and calling all those critical results. You could imagine how you must be feeling at 3:00 AM in the, in your night shift, where you have so much work to do. And sometimes we were even asked to handle the bacteriology side as well when there was no support of a lab assistant during night.”*



*FG-95: “When you’re, when, during our testing, we finish, we usually try to finish the whole process and not leave it to someone else because it’s just very difficult to hand off sometimes. So, um, unpaid overtime, meaning that a lot of times to finish it before the shift ended, we wouldn’t take our break times. So, we would just skip our breaks entirely or we would just take an extra 10 min to finish resulting the samples. And it wasn’t something that was forced, it was just that a lot of people, it was just so busy that you just want to finish your work and go home and have no conscious about it and not take that work home. And then you just didn’t want to leave thinking that, um, you know, that you didn’t hand off properly, it was rushed, and it was just a lot of just emotional kind of, you didn’t want to leave home thinking about work.”*


Participants expressed that they had felt guilty when considering sick leaves or absences, as they were aware of the already-existing staffing challenges. They were concerned about the impact their absence may have on the remaining staff and leaving their colleagues with an even greater burden. This fear of overburdening their colleagues created a sense of obligation to remain at work, even when they were unwell and needed some time off.


*FG-89: “…You would see on the schedule, um, that they’re working short that day and then yeah so, I had uh, yeah, even, even myself like, I would, I would always like triple think even, even trying for one, because you just know that it wouldn’t be possible. Yeah, and even some people who are genuinely sick, like they would feel so guilty about calling in sick to work because they just know it’s, it’s just gonna mean like, an even worse day for everyone.”*


Given the sense of duty to support their colleagues, some retired and soon-to-retire MLTs were reportedly compelled to continue working to alleviate the staff shortages. Our results revealed that those who had planned to retire chose to postpone their plans, as they were cognizant that their absence would have left an additional strain on their colleagues. Some participants also reported that they knew of retired co-workers who had returned to work to fill staffing gaps and support the workforce. 


*FG-101: “People that had like their retirement plan, like planned out, they had their dinner ready they were like, “yeah, I’m gone.” COVID kind of hit, and they were like, “I’ll stay. I will, like, stay, until this is all figured out.” We had people come back, like it was the opposite here, people that were retiring stayed on to see us through. They’re still working to make sure that we, like, get through this.”*


### 3.4. Revealing the Realities of the Hidden, yet Indispensable Role of MLTs in Predominantly Non-Patient-Facing Roles

#### 3.4.1. Limited Understanding and Recognition of the Importance and Roles of MLTs in Healthcare

Despite the invaluable roles of MLTs in the healthcare system, their efforts and contributions go unnoticed. Participants expressed their belief that there was a lack of understanding of their roles by those outside of the laboratory. As MLTs are typically non-patient-facing, participants noted the limited public awareness and understanding of their profession. Participants expressed the belief that they felt forgotten in healthcare in comparison to other healthcare workers and that they were perceived as a ‘magic black box’ where samples are automatically analyzed and interpreted. They noted that individuals outside the laboratory failed to recognize the complexities within the laboratory. Participants emphasized that the lack of public awareness and understanding of their profession may be attributed to the non-patient-facing nature of their work. They explained that the limited visibility of MLTs may have contributed to the lack of adequate funding for staffing, as the attention and allocation of resources tended to be directed towards more visible areas of healthcare. Participants also expressed their sense of a lack of understanding of laboratory operations among other healthcare workers, in demanding faster turnaround times and expecting immediate responses. This lack of understanding led to challenges in communicating and collaborating with other healthcare professionals. 


*FG-53: “Some of the stressors that I’ve experienced in this profession, for the 27 odd years, primarily boils down to a lack of either understanding and or respect for the work that actually is accomplished within the laboratory. People outside of the lab kind of view us as that magic black box, samples will go, come, go in. Magic happens. Numbers come out, values come out, but what happens in between is magic… I think that’s the single biggest challenge I’ve experienced in my long career…We’re not in the front, to, for the world to see… we’re hidden, and I think that has a lot to do with why we’re not funded to the levels required to achieve the staffing that’s required to make it possible for us to feel not spread thin.”*



*FG-96: “I would say our common stressor is, uh, communication and dealing with other health care professionals who may or may not understand, um, really how our profession works and the, the sort of rules the, um, standards that we need to follow.”*


#### 3.4.2. Lack of Compensation despite Increased Work Demands and Responsibilities

Despite their contributions during the pandemic, participants felt that their work was unrecognized and unappreciated. MLTs found themselves with increased work demands and responsibilities without receiving any additional compensation. Participants expressed their frustrations, as they were tasked to assume more responsibilities without receiving any additional compensation for the increased workload. 


*FG-49: “…you’re asked to take on more responsibilities without actually any additional compensation.”*


Several participants reported they did not receive ‘pandemic pay’ or any additional financial compensation for their efforts and sacrifices during the crisis. This lack of recognition and compensation resulted in dissatisfaction among MLTs, which impacted their work morale and made them question the value of their work. Participants perceived that their efforts were underappreciated and that they were not given the same recognition as other healthcare workers, leading them to question the significance of their roles. They expressed disappointment with the unequal treatment, as they believed that their contributions were just as important, and equal to those of other healthcare workers.


*FG-84: “And then, um, I know then in Ontario, one of the stressors was they did like a pandemic pay for like nurses and people that work in the, like the dietary people, the secretaries, they were getting a pandemic pay, but then the MLTs and respiratory technologist and diagnostic imaging, they didn’t get that cause they weren’t first, first line workers…Like, we were like right there, but it was never acknowledged.”*



*FG-53: “And then, the lack of pandemic pay hit. And I am sorry, every lab tech in this thing has cried during this pandemic. And then to have that pandemic pay slap in the face hurt. It was like, so it really doesn’t matter what we do, so maybe we should just stop doing it. So, there was that.”*


#### 3.4.3. Lack of Management Support

MLTs also expressed a belief that they had experienced a lack of support from management during the pandemic. Participants reported that despite voicing their concerns regarding the staffing issues and needing more MLTs, they felt their pleas had gone unheard. Participants expressed their frustrations with their managements’ inaction to their concerns about staffing shortage, which negatively impacted work morale. Participants also highlighted the impact of hearing words of acknowledgement, even when immediate actions to address these concerns might not have been feasible. They expressed their sense that simply acknowledging their concerns and reassuring them that efforts would be made to support them could have helped to boost morale and job satisfaction. 


*FG-51: “There was also that sense of management not hearing us where we were constantly banging on the fact that we were really short [and] we need to hire more people. We need to do something about this and, uh, almost the feeling like the concerns with being brushed off they weren’t even really being acknowledged. Um, and then sometimes even if management can’t do anything, if they say, ‘But our, our hands are tied, we can’t really do anything about it, but we see you, we’ll do what we can.’ Sometimes even hearing that helps, but our hospital didn’t really do that. So that kind of also affected morale.”*


Participants also reported a lack of regard and respect for their personal lives. One participant recounted the time they had requested a leave of absence due to a family emergency; however, their request was denied due to staffing limitations. The participant expressed dismay and dissatisfaction at the lack of understanding and consideration for personal circumstances. Participants felt they were not supported in their time of need and felt undervalued as an employee. 


*FG-56: “I also had a family emergency during, I think, wave three of the pandemic and I wasn’t allowed to take time off work to deal with that. So, what, management is fully aware of what I was going through, but I was told we’re short-staffed.”*


#### 3.4.4. Challenges Accessing Mental Health Supports and Services for MLTs

Participants also highlighted the challenges of accessing mental health support and services during the pandemic. Participants noted the importance of mental health during the pandemic. Some participants reported a lack of available mental health services. Other participants reported that despite the availability of these services, they were not accessible to them, as they did not have the time to attend them outside of work.


*FG-99: “…it’s usually really difficult, like, given like how short we are, not everyone actually has the time outside the work to attend these, uh, sessions on their personal time.”*


### 3.5. Leveraging Insights from the COVID-19 Pandemic to Enhance Healthcare Practices and Preparedness

#### 3.5.1. Recognizing One’s Own Limitations and Establishing Boundaries for a Healthy Work-Life Balance

Participants recognized the importance of prioritizing their well-being and setting boundaries in the workplace. Working throughout the pandemic, participants gained valuable insights into their own capacities and limitations. The participants recognized the significance of self-care and realized they must prioritize their own health and well-being. Participants noted that by placing themselves first, they are better equipped to care for patients and fulfill their professional duties effectively. They indicated that if they are healthy, both physically and mentally, they can be more productive and are better able to handle the demands of their work.


*FG-99: “Um, just knowing what my own limits are and just really know how I am and what I can take on is that, you know, as a medical lab tech. And also, how to handle the various stressors either at work or at home, so that, that’s really became, um, I would say the highlight of what I’ve learned throughout this whole pandemic period, to figure out, uh, more of my own, uh, yeah, mental capacities.”*



*FG-94: “…I learned to set up boundaries for myself. I was more of like a yes person before, and even if there were things I wouldn’t [be] uncomfortable with or I didn’t agree with, I would just kind of think of it as a compromise and just let things be. But to me this was a really big deal and I learned to kind of stand up for myself and my family and set up boundaries which, uh, some people were not used to, but it was, uh, it was kind of like, uh, I got a little stronger in a sense, I guess. So, that was, that was kind of the lesson I learned in kind of the good thing that came out of it.”*



*FG-107: “And also, you know, in order to serve the citizens of Manitoba, you know, I, I need, I needed to, to remember that I need to place myself, ensure that I’m placing myself first and I wasn’t doing that… I was just always thinking about somebody else, but it got to the point where I was putting other people first and then it was having its toll on me.”*


#### 3.5.2. Integrating Leadership and Communication Strategies in Preparedness Plans for Effective Public-Health Emergency Response

Participants noted the importance of having strong leadership and effective communication for enhancing preparedness and response efforts during a public-health crisis. They recognized that clear and consistent communication is paramount in such situations. Participants underscored that timely and coordinated communication is crucial during public-health emergencies. They noted that effective communication channels should be established to disseminate accurate and up-to-date information, guidelines, and protocols to reduce confusion among staff and ensure that everyone is well-informed. 


*FG-82: “I think a major thing and it’s a tough thing to be able to do, given that, you know, responding to a pandemic is ever-changing but consistent messaging and consistent, um, like a consistent approach from leadership, uh, in enforcing rules or even determining what the rules are, and very clearly and consistently communicating those rules would have reduced some stress.”*


Participants also highlighted the critical role of strong leadership during a public-health crisis. They emphasized the need for leaders who can provide guidance and support, ensuring that each member of the team understands their role and carries out their responsibilities accordingly. They highlighted their belief that leaders should be responsive to the needs and concerns of their teams and foster an environment where individuals feel valued and supported. By integrating communication and leadership strategies into preparedness plans, healthcare organizations can enhance their response capacities as to future public-health emergencies.


*FG-81: “…in terms of preparedness, having a plan for future pandemics, I think it’d also be important to emphasize the role of leadership. Who is doing what and where? Who’s taking care of these staff? Who’s taking care of this patient, for that patient? During the pandemic in my region, everybody was their own leader. We were left to a lot of our own devices… So, in the future, knowing that we have certain people we can look up to, and knowing that we can go to them to find what our roles are, know what their roles are, know what we need to do as individuals would better help not just us understand things, help our morale, help our stress, but give the patients better care.”*



*FG-80: “…communication was probably the number one thing that, sort of, wasn’t wonderful for this pandemic and, and also just leadership in general. Like, I think people need to feel like they’re a part of a team, but also that they have someone leading them who actually, I mean, I don’t know what one knew what was going on, but it was, it was, everything was so uncertain. And if they could, like, make clear plans…I think if leadership really made stronger protocols and actually stood behind them, it, it gives you sort of a feeling of where you’re supposed to be because everything was really up in the air, and we didn’t have sort of any kind of leg to stand on.”*


#### 3.5.3. Enhancing Healthcare Quality and Efficiency through Electronic Health-Record Integration

Participants expressed their belief that having an integrated electronic health-record system would improve communication among healthcare providers. By integrating patient information into a centralized electronic system, healthcare providers can access patient information, including laboratory results, more efficiently. Implementing an electronic health-record system can help streamline workflow and increase efficiency, as less time is spent on administrative tasks, contributing to increased productivity among healthcare professionals. The adoption of such a system can also facilitate continuity of care for patients and help minimize errors, which will, overall, enhance patient care. 


*FG-82: “So, I think, kind of a foundational piece that could be improved across the board is the electronic health record. Um, if that was really seamless with our test results going in and, um, being accessible as soon as they’re ready and, and all of that, I feel like communication between healthcare providers could be improved, patient care could be improved, you know, our workload could be reduced if that was really seamless, just across the board, that would be an improvement. And that could be, you know, a great source of data for epidemiology that would be a little bit more consistent if it was, uh, yeah, more cohesive across Canada or even within provinces, um, between health authorities and between the private and public healthcare, um, like, uh, private lab companies, results being able to come into the EHR in a hospital setting in a simple way would make things simpler.”*


## 4. Discussion

The purpose of this qualitative study was to identify and understand the stressors of MLTs working during the COVID-19 pandemic and to identify lessons learned from the pandemic. Research on this population was scant even before the pandemic, suggesting the need for further examination. To our knowledge, this is the first national qualitative study exploring the experiences of MLTs during the COVID-19 pandemic. The results revealed several factors that contributed to the stress experienced by MLTs during the COVID-19 pandemic, including exposure to the virus, high workload, extended work hours, lack of recognition and compensation, and lack of management support. These findings are similar to those of a provincial study conducted on medical laboratory professionals in Ontario, Canada by Gohar and Nowrouzi-Kia [25]. This suggests MLT challenges are not unique to one province and are present at a national level.

### 4.1. Pandemic Stressors

The first theme that emerged from the study was: unexpected challenges of MLTs navigating through the uncertainties of an ever-evolving pandemic. The pandemic inherently changed the landscape of the healthcare system, including laboratories, and presented new challenges to MLTs. Our findings indicate that the evolving nature of the pandemic and limited understanding of the virus caused uncertainty and fear among MLTs. The frequently changing protocols and procedures during the pandemic added to the stress and anxiety felt by MLTs, as they had to stay updated with new information and adapt quickly to the rapid changes. This fear of the unknown during the pandemic was seen among healthcare workers in previous studies [33,34]. Due to the highly contagious nature of the COVID-19 virus, MLTs who worked directly with potentially infected patient samples described their fear of contracting the virus and the risk of transmitting it to their families. Similar studies show the fear of infection and transmission to patients and household members as a significant stressor among healthcare workers [6,33,35]. During disease outbreaks, healthcare workers face a substantially higher risk of infection compared to the general population, causing considerable stress and anxiety [35,36]. In an Ontario study, healthcare workers had a disproportionately higher infection rate compared to non-healthcare workers [37]. Our findings indicate that MLTs worked with limited protection due to PPE supply shortages, increasing their susceptibility to the virus, which heightened their anxiety about their safety. Many healthcare workers experienced this lack of adequate PPE during the pandemic, which was a source of fear and frustration [38]. This issue highlights the need for proper PPE allocation and distribution to ensure healthcare workers’ health and safety. Participants also reported that the shortages of essential testing supplies and equipment amid the high testing demands elevated their stress levels. The pandemic affected the demand and supply of healthcare supplies and equipment, causing disruptions to the supply chain [39]. Participants noted that the lack of necessary supplies impeded their work, leading to delays in testing and diagnoses, compromising patient care.

The second theme of this study outlines the implications of staff shortage for MLTs’ well-being and patient care. Our findings indicate that the increased testing demands during the pandemic resulted in increased workload and longer working hours for MLTs. The pressure to deliver accurate and timely results placed immense strain on MLTs, leading to exhaustion and burnout. This finding is consistent with the results of an Ontario study which shows a high prevalence of burnout among MLTs [17]. High levels of burnout among healthcare workers during COVID-19 have consistently been reported in previous studies [40,41,42,43,44,45]. A study showed that longer working time increases the risk of occupational burnout [46]. Long working hours have been linked with occupational health problems such as sleep disturbances, anxiety, depression, and cardiovascular diseases [42,47,48,49,50,51]. 

Prolonged burnout can lead to attrition, further aggravating the staffing shortage. Our findings indicate that burnout from the increased workload caused many MLTs to leave the profession and retire early. Recent national data show that burnout (63.2%) emerged as the most commonly reported reason among healthcare workers considering leaving or changing jobs [52]. Moreover, our results offer the insight that less-experienced MLTs faced overwhelming pressures to keep up with the workload during the pandemic, causing them to leave the workforce. Participants noted that due to the staff shortage, there was limited capacity to adequately train new MLTs, leaving them unable to cope with the demands of their jobs. Similarly, a study exploring burnout among nurses revealed that early-career nurses reported higher rates of burnout, fatigue, and emotional exhaustion, with a large proportion reporting that they planned to leave the profession within the next five years [53]. Likewise, evidence from a Turkish study suggests that burnout was the main predictor of turnover intention among healthcare workers during the COVID-19 pandemic [54]. Also, data from a nationwide survey show that high levels of burnout and intent to leave their jobs are linked to high workload across various healthcare roles during the time of the COVID-19 pandemic [55]. 

Job burnout also contributed to increased laboratory errors, which can have serious consequences for patient care and safety. This finding mirrors previous research linking burnout to patient safety [56]. Previous studies on burnout among doctors have been linked to suboptimal patient care and increased medical errors [57,58]. Another study showed a higher frequency of medical errors among healthcare workers with poor sleep quality during the pandemic [47]. When healthcare workers face burnout, their ability to deliver patient care may be compromised, resulting in reduced productivity and increased incidence of errors. 

The study’s findings also revealed that MLTs experienced moral distress during the pandemic. Previous studies on nurses show that moral distress is a significant predictor of turnover intention [59,60]. The findings indicate that MLTs felt guilty for taking breaks or leaves of absence from work, fearing that their absences would overburden their colleagues. Previous studies on physicians and nurses have demonstrated that presenteeism, or attending work while unwell, is related to reduced productivity and can result in significant financial costs [61,62]. Presenteeism among healthcare workers has also been shown to have negative impacts on the quality of care and patient safety. In a study on nurse presenteeism, the results show an increased incidence of medication errors and patient falls [63]. Similarly, a study on presenteeism in pharmacists revealed that pharmacists who attended work while unwell made significantly more minor errors and serious mistakes than those who did not [64]. Another study shows that nurse presenteeism resulted in disease transmission to patients [65]. These findings suggest that working while unwell can potentially compromise patient safety, highlighting the importance of ensuring the well-being of employees.

While there are commonalities between the first two themes and other healthcare workers, our third theme identifies stressors that are unique to MLTs as typically non-patient-facing healthcare workers operating in isolated laboratories. Participants indicated that those outside the laboratory had limited understanding of the roles of MLTs and of the laboratories’ operations. This lack of understanding contributed to the MLTs’ stress, as they were expected to increase their workloads with limited human and laboratory resources.

This lack of recognition contributed to stress and job dissatisfaction among MLTs. Compared to other healthcare workers, there was a lack of public recognition for the work of MLTs, which left participants feeling undervalued and demotivated. A study on nurses shows that recognizing and appreciating the work of employees promotes job satisfaction [66]. Recognizing MLTs’ contributions to the healthcare system and giving meaning to their work can improve job satisfaction and retention. Our findings also indicate that some MLTs did not receive the ‘pandemic pay’ which was designed to support frontline workers fighting COVID-19. While this incentive was well-intended, it led to organizational injustice among healthcare workers, including MLTs who did not receive such compensation. Researchers who examined healthcare workers found that organizational injustice, or the unfair treatment experienced by employees in their workplace, is a significant predictor of employees’ organizational commitment, job satisfaction, and turnover intention [67,68]. Participants in the study cited the lack of compensation as a factor that caused some to consider leaving their positions or seeking alternative employment. This finding aligns with a study examining turnover rates among MLTs which found low salaries and low annual raises are the primary factors contributing to turnover [69]. 

Our findings also indicate a lack of management support, leading to decreased morale among MLTs. Poor leadership can lead to employees feeling undervalued and unsupported, affecting job satisfaction and motivation [70,71,72]. Results from a meta-analysis examining predictors of sickness absence among nursing personnel revealed that low work-support increased the likelihood of sick leave [73]. Organizations should consider investing in leadership training programs that educate leaders on the ways to support and empower their team members. Moreover, leaders should adopt participative leadership styles that engage team members in the decision-making process [74]. This supportive approach boosts morale, job satisfaction, commitment, and retention while enhancing productivity and improving patient care [75]. By involving team members in decision-making processes and decisions, managers promote a sense of empowerment. This collaborative approach creates a culture of trust, boosts morale, and enhances overall productivity within the team.

### 4.2. Staff Shortage: The Root of All Evil

Staff shortage is a concerning and persisting issue in the healthcare sector across Canada and worldwide. Our results revealed that staff shortage, when combined with increased work demands, leads to poorer health outcomes, exacerbating the staff shortages due to sick leaves or employees leaving the profession. This is consistent with a model proposed by Gohar and colleagues [76], depicting the cyclical process of staff shortage in healthcare due to direct and indirect factors during the pandemic (Figure 2). Direct factors refer to situations that automatically cause staff shortages due to events, such as illnesses or accidents occurring either in the workplace or outside the workplace. Indirect factors are not as easily detected; however, they can be traced over time. Beginning with staff shortage, the initial antecedent (i.e., level 1), existing healthcare workers face higher work demands (i.e., the moderator). Given situations including personal factors (e.g., lifestyle) and/or organizational factors (i.e., level 2) such as poor management, healthcare workers eventually develop physical or psychological conditions (i.e., level 3), forcing them to go on sick leave, leave the organization, or leave the profession. This model can be used by administrators and policymakers to consider their employees’ physical and psychological well-being, along with the organization’s level of support and policies that may contribute to solutions for staff shortage. 

In addition to the existing healthcare staff shortage, there is an unprecedented concern for Canada’s laboratory workforce. According to CIHI [19], there are approximately 51 MLTs per 100,000 population. This is particularly concerning, as Canada’s population is growing faster and growing older than ever. As of June 2023, Canada’s population had reached 40 million [77]. In 2022, the population grew by 1,050,110, marking the highest annual population growth rate since 1957 [77]. As an added caveat, the Canadian population is aging, with more Canadians aged 65 and above than under 15, which translates to more medical needs. One in five of the working-age population (aged 15 to 64) is nearing retirement, contributing to the labor shortages across the country [52]. With Canada’s growing and aging population and half of the MLTs in the workforce nearing retirement [20], there is a need to increase laboratory capacity by improving retention among existing MLTs and recruiting new MLTs.

In response to the critical shortage of MLTs, recent government initiatives have been implemented. In Ontario, steps have been taken to increase laboratory capacity by integrating internationally educated MLTs into the workforce and expanding the number of seats per cohort [78]. Also, a new three-year online program has been introduced in Nova Scotia to train up to 40 additional MLTs in the province [79]. These steps are promising; however, various efforts are required to sustain the profession, which would, ultimately, preserve the Canadian healthcare system, as laboratory services are critical in diagnostics and treatment.

### 4.3. Recommendations

Our results revealed that MLTs recognized the importance of establishing boundaries in the workplace. A clear work–life balance increases job satisfaction, which promotes employee commitment to the organization [80]. Specifically, by having clear expectations of working hours, despite the high volume of work, the employee is less likely to leave the organization, reducing job turnover and thus facilitating better health services. Participants emphasized the importance of prioritizing their mental health and well-being, as this impacts their ability to deliver care. Healthcare organizations should prioritize a positive culture, offering counseling, mental health programs, and stress management to combat workplace stress. Investing in employee well-being enhances productivity, benefiting patient care and healthcare efficiency. Perceived organizational support reduces burnout risk, improves patient care, and boosts resilience [81], which are important factors during health crises. 

Our findings also indicate the need for strong leadership and clear communication during pandemics. MLTs reported a lack of clear guidance and communication from their leaders during the pandemic, which made it difficult for them to understand their roles and responsibilities. This finding is consistent with other studies that show healthcare workers navigating the pandemic with ambiguity due to the lack of communication and leadership [82]. Consistent communication and effective leadership lead to lower perceptions of stress and burnout among emergency frontline workers [83]. Person-centered leadership like transformative leadership fosters a supportive work environment, increasing productivity and organizational commitment [70,71,72]. A supportive work environment includes actively promoting open communication and collaboration among team members. Transparent and open communication increases resilience in healthcare workers [83,84]. Also, recognizing employees’ efforts increases motivation and job satisfaction, reducing turnover intention [85]. 

Effective communication for service facilitation was also reported as a method for improving the healthcare system. Participants reported that an electronic health record would improve communication among healthcare providers, lessen their workloads, streamline laboratory processes, minimize laboratory errors, and ultimately improve patient safety. In a study examining job satisfaction and digital healthcare, the authors uncovered the fact that physicians who frequently use digital health experienced greater job satisfaction in comparison to those who did not adopt such technologies [86]. We recommend that electronic health records be streamlined across all healthcare settings and between healthcare workers in order to streamline services more effectively. 

To increase the recruitment and retention of MLTs, we recommend increasing awareness and public education about the vital role of MLTs and their work in the healthcare system. Increasing the visibility of the profession can attract more individuals who wish to pursue laboratory medicine and can also help increase funding to support the staffing shortage. Further, raising awareness of the profession will lead to more realistic expectations of their work and ensure that their contributions are duly recognized. We also recommend that when offering incentives to healthcare workers, they should be offered more equitably to minimize organizational injustice. 

### 4.4. Limitations

An inherent limitation of qualitative research is the difficulty in generalizing the results. Despite the small sample size, our findings were consistent across different provinces. Furthermore, provincial [17,21,25] and national data [3,52] present similar concerns. As such, while our study may lack generalizability, this research should be viewed as explanatory, providing detail and context to existing quantitative data, which is valuable for researchers and for policymakers. Furthermore, despite our recruitment efforts, it was challenging to recruit from all provinces. This could, in part, be due to the notable stressors discussed in the study, as they affected participation. Nevertheless, from a geographical context, we managed to capture experiences from the western, eastern, and central regions.

## 5. Conclusions

Working behind the scenes, MLTs are often unseen, despite their crucial role in the healthcare system. The pandemic highlighted their importance, as they were at the forefront of the response to COVID-19. This qualitative study aimed to identify and understand the stressors of MLTs during the pandemic, and to determine lessons learned for the improvement of employee wellness and healthcare provision. The results identified key themes, including (1) unexpected challenges navigating through the uncertainties of an ever-evolving pandemic; (2) implications of staff shortage for MLTs’ well-being and the quality of patient care; (3) revealing the realities of the hidden, yet indispensable role of MLTs in predominantly non-patient-facing roles; with the fourth theme highlighting the lessons learned. Our recommendations emphasize the need to address the staff shortages in this population due to commonly known issues in healthcare such as high work demands and organizational factors, in addition to Canada-specific factors, including Canada’s aging and growing population and a substantial proportion of MLTs nearing retirement. By implementing these recommendations, healthcare organizations can better support MLTs and ensure the sustainability of the laboratory workforce, ultimately enhancing the delivery of quality medical laboratory services in Canada. The findings of this study could be used by policymakers to strengthen the healthcare system’s capacity to respond to future public health emergencies. 

## Figures and Tables

**Figure 1 healthcare-11-02736-f001:**
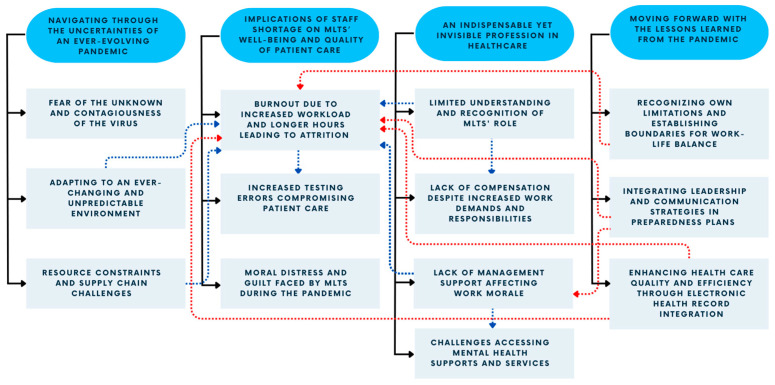
Thematic map depicting the stressors and the lessons learned during the pandemic. The dark blue ovals represent the themes, and the light blue rectangles represent the subthemes. The solid black lines represent the link between the themes and subthemes. The dotted blue lines represent a positive (or increased) effect, and the dotted red lines represent a negative (or decreased) effect between subthemes.

**Figure 2 healthcare-11-02736-f002:**
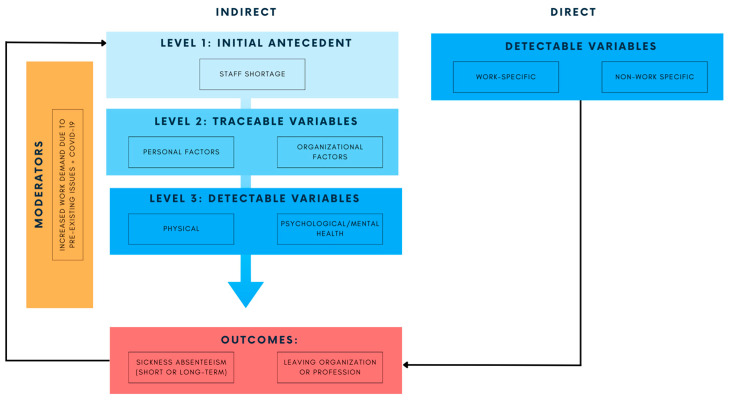
Model depicting the cyclical process involving staff shortage in healthcare [76].

**Table 1 healthcare-11-02736-t001:** Participants’ demographic information.

Gender	Age (Years)	Province	Marital Status	Location	Work Setting	Total Length of Practice	Length of Practice at Current Location
Men = 4 Women = 23	21–25 = 3 26–30 = 4 31–35 = 5 36–40 = 8 41–45 = 2 46–50 = 1 51–55 = 4 (M = 37, SD = 9.3)	Ontario = 9 British Columbia = 8 Alberta = 6 Nova Scotia = 2 Manitoba = 1 Newfoundland and Labrador = 1	Single = 11 Common-Law = 4 Married = 12	Urban = 19 Rural = 4 Mixed = 4	Hospital = 19 Private laboratories = 2 Provincial laboratory = 1 Clinic = 1 Community center = 1 Non-profit organization = 1 Manufacturing/distribution = 2	Min. = 2 years Max. = 35 years (M = 10, SD = 8.7)	Min. = 2 months Max. = 26 years (M = 6, SD = 6.6)

**Table 2 healthcare-11-02736-t002:** Themes and subthemes.

Themes	Subthemes
Theme 1: Unexpected challenges navigating through the uncertainties of an ever-evolving pandemic	Subtheme 1: Fear of the unknown and contagiousness of the virus Subtheme 2: Adapting to an ever-changing and unpredictable environment Subtheme 3: Resource constraints and supply chain challenges
Theme 2: Implications of staff shortage for the well-being of MLTs and the quality of patient care	Subtheme 1: Burnout due to increased workload and prolonged work hours leading to attrition Subtheme 2: Increased testing errors compromising patient care Subtheme 3: Moral distress and guilt faced by MLTs during the pandemic
Theme 3: Revealing the realities of the hidden, yet indispensable MLTs in predominantly non-patient-facing roles	Subtheme 1: Limited understanding and recognition of the importance and role of MLTs in healthcare Subtheme 2: Lack of compensation despite increased work demands and responsibilities Subtheme 3: Lack of management affecting work morale Subtheme 4: Lack of available mental health supports and services for MLTs
Theme 4: Leveraging insights from the COVID-19 pandemic to enhance healthcare practices and preparedness	Subtheme 1: Recognizing one’s own limitations and establishing boundaries for a healthy work-life balance Subtheme 2: Integrating leadership and communication strategies in preparedness plans for effective public health emergency response Subtheme 3: Enhancing healthcare quality and efficiency through electronic health record integration

## Data Availability

The data supporting the findings of this study are not available due to ethical restrictions.

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
