# Peer review of "Hidden and Understaffed: Exploring Canadian Medical Laboratory Technologists’ Pandemic Stressors and Lessons Learned"

_healthcare, 2023, doi:10.3390/healthcare11202736_

Round 1

Reviewer 1 Report

Dear Authors,

the text concerns a noteworthy aspect of the work "in the shadow" of laboratory personnel. It starts very smoothly, even engaging the reader. But then things get complicated, because the next steps in the research procedure are unclear:

1. how were people selected to each of focus group?

1. there are no attachments with questions according to which the focus group study was conducted. However, the text contains information that semi-structured questions were used.

We used a semi-structured interviewing method and guided the discussion with open-ended questions (Supplementary File). Follow-up questions were also added as necessary.

However, in the next lines, five general questions are given.

This part of the work is not clear.

2. the question also arises: how were the codes received in the program Quirkos used? In the results section we basically have a set of quotes....admittedly interesting.

3. Was it possible to identify regional specifics during the conversations? there is little or nothing on this topic in the text.

Line 170-183   how it was emerged from Authors data? Any explanation is needed.

Four key themes emerged from our data: (1) navigating through the uncertainties of an ever-evolving pandemic; (2) implications of staff shortage on MLTs’ well-being and quality of patient care; (3) an indispensable yet invisible profession in healthcare; and (4) moving forward with the lessons learned from the COVID-19 pandemic (Table 2). Figure 1 shows the thematic map outlining the relationship between themes and sub-themes.

In the Discussion section, too much space is taken up by repetitions of content from the previous chapter. Here it is enough to indicate the discussed aspects with just a short term/code.

 After making clarifying changes to the text, the article will be worth publishing with the hope that it will have a scientific impact.

The text is written correctly, but there are repetitions, e.g. in Data Analysis (we used it several times).

Author Response

Thank you, please see attached. 

Reviewer 2 Report

This is a well-written paper with rich insights into the experience of an under-studied category of healthcare workers who are critical (as the paper's conclusions support) to the functioning of healthcare systems, both during and beyond crises like COVID. I think readers will be gripped by the human story told in this paper, and administrators and researchers will find actionable insights in the findings and recommendations presented. 

The study design is well presented and appropriate, and while the sample is small the findings are consistent with other insights from the literature (which is appropriately and thoughtfully cited). The implications drawn do not overstate the strength of the findings or the potential of the design, but are realistic.

I have minor suggestions for improvement and/or some questions the authors may wish to consider in a revised paper:

- Who is the "we" of the research team? A little bit (just a couple phrases or a sentence or two) of reflexive description of the study team would be helpful:  the professional/disciplinary lenses they applied to the analysis of the data, and the potential ways this influenced participants' interactions with them (a focus group led by MLTs, for example, might get different results from one led by people with no "insider" knowledge of the job).

- The study demographics collected gender but not race/ethnicity. This is not anything you can change, but I am curious 1) why gender info was collected but not considered in the analysis, and 2) why race/ethnicity info was not collected. You might wish to provide a rationale for these choices. 

- The paper states that thematic saturation was used to determine when the researchers stopped conducting focus groups. The authors would be doing a service to the field of qualitative health research if they would explain how their analytic process was integrated with the data collection timeline, so that they knew they had reached saturation. As written right now, one could read the analysis/coding as all occurring after data collection was finished (which often is the case), and this would be inconsistent with using saturation to determine when to finish sampling. I encourage the authors to give more detail here about the timeline integration of analysis and data collection.

- A really nice feature of the paper, in my opinion, is its use of quotes in the body of the text, creating a rich and continuous narrative, as well as the use of a summarizing table of themes and subthemes, and a figure showing the relationships between the subthemes. These are all different ways that different readers will appreciate making sense of the findings. A couple comments/suggestions on the figure:

1) are the relationships represented by the arrows theorized by the authors, or were they commented on/offered by participants themselves. This links to another comment/suggestion I have : in the methods description of the analysis, it would be good to say more to what extent the team's findings were created using an interpretive approach versus a purely descriptive one, reporting what participants said. Some of the links represented in the figure - notably the link from effective communication to having an effect on management practices that in turn affected morale - make sense, but are directly supported by quotes. It would be good to know how much the "sense" made of the data was driven by participants' own sensemaking and how much from the researchers (both are legitimate, but the paper would be improved by being clearer on this point). 

2) The language around "positive" (blue arrows) and "negative" (red arrows) effect represented by dotted lines in confusing, since positive can mean both "good" and "increased." Positive or negative "association" might be less confusing, but that word implies statistical relationships for many readers.  In particular: "burnout" is linked to "testing errors" with a blue arrow (more burnout increases/leads to more errors - makes sense), but testing errors are linked to burnout with a red arrow - so more testing errors are related to less burnout? Or is the figure trying to say that reduced testing errors would lead to reduced burnout? Also, if that were the case, that is more of a theorized model than a figure of the themes in the finding. The main point is: some of the language/presentation in the figure could be clearer.

- Finally, in the discussion of recommendations: given that the findings show that unsupportive management was a significant problem participants experienced, how do the authors think that organizations and individual managers will actually, truly be moved to support MLTs in practicing healthy boundaries? There's more of a challenge here than is suggested by simply saying that healthcare systems "must provide support and resources...for self care" (p21), since the findings suggest that healthcare systems don't (or haven't) prioritize these resources. How do system initiatives get managers to readily give their direct reports time and genuine support to take care of themselves? It would be good to hear ideas the authors have for how to crack this tough nut, and perhaps acknowledge a bit more just how much transformation might be needed to get systems to do what they "should."

Author Response

Thank you, please see attached. 

Reviewer 3 Report

This is an excellent qualitative study.  I have one comment.  The overall themes are written like categories not themes.  Themes should have some substance not simply statements that organize subthemes.  For example, the first theme could be stated as "Unexpected challenges navigating through the uncertainties of an ever-evolving psndemic."

Author Response

Thank you, please see attached.
